# Unsupervised Learning of State Representations for Multiple Tasks

**Antonin Raffin**[1]
École Nationale Supérieure de Techniques Avancées (ENSTA-ParisTech), Paris, France
`antonin.raffin@ensta-paristech.fr`

**Sebastian Höfer**[1]**, Rico Jonschkowski & Oliver Brock**
Robotics and Biology Laboratory, Technische Universität Berlin, Germany
`{sebastian.hoefer,rico.jonschkowski,oliver.brock}@tu-berlin.de`

**Freek Stulp**
Robotics and Mechatronics Center, German Aerospace Center (DLR), Wessling, Germany
`freek.stulp@dlr.de`

## ABSTRACT

We present an approach for learning state representations in multi-task reinforcement learning. Our method learns multiple low-dimensional state representations from raw observations in an unsupervised fashion, without any knowledge of which task is executed, nor of the number of tasks involved. The method is based on a gated neural network architecture, trained with an extension of the *learning with robotic priors* objective. In simulated experiments, we show that our method is able to learn better state representations for reinforcement learning, and we analyze why and when it manages to do so.

## 1 INTRODUCTION

In many reinforcement learning problems, the agent has to solve a variety of different tasks to fulfill its overall goal. A common approach to this problem is to learn a *single policy* for the whole problem, and leave the decomposition of the problem into subtasks to the learner. In many cases, this approach is successful (Mnih et al., 2015; Zahavy et al., 2016), but it comes at the expense of requiring large amounts of training data. Alternatively, *multiple policies* dedicated to different subtasks can be learned. This, however, requires prior knowledge about how the overal problem decomposes into subtasks. More-

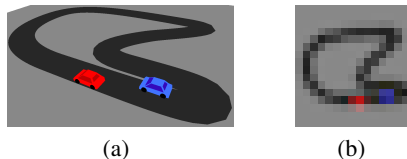

(a)                              (b)

Figure 1: Slot car racing – the agent has learn how to drive any of the cars as far as possible (left), based on its raw observations (right).

over, it can run into the same issue of requiring large amounts of data, because the subtasks might overlap and thus afford shared computation to solve them.

A common approach to address overlapping problems is *multi-task learning* (Caruana, 1997): by learning a single policy with different subgoals, knowledge between the different tasks can be transferred. This not only allows to learn a compact representation more efficiently, but also improves the agent's performance on all the individual subtasks (Rusu et al., 2016).

Multi-task learning, however, faces two problems: it requires the decomposition of the overall problem into subtasks to be given. Moreover, it is not applicable if the subtasks are unrelated, and are better solved without sharing computation. In this case, the single-policy approach results in an agent that does not perform well on any of the individual tasks (Stulp et al., 2014) or that unlearns

---

[1]The first two authors contributed equally to this work.

the successful strategy for one subtasks once it switches to another one, an issue known as catastrophic forgetting (McCloskey & Cohen, 1989).

In this work, we address the problem of identifying and isolating individual unrelated subtasks, and learning multiple separate policies in an unsupervised way. To that end, we present MT-LRP, an algorithm for learning state representations for **m**ultiple **t**asks by **l**earning with **r**obotic **p**riors. MT-LRP is able to acquire different low-dimensional state representations for multiple tasks in an unsupervised fashion. Importantly, MT-LRP does not require knowledge about which task is executed at a given time or about the number of tasks involved. The representations learned with MT-LRP enable the use of standard reinforcement learning methods to compute effective policies from few data.

As explained before, our approach is orthogonal to the classical multi-task learning approach, and constitutes a problem of its own right due to the issues of underperformance and catastrophic forgetting. Therefore, we disregard the shared knowledge problem in this paper. However, any complete reinforcement learning system will need to combine both flavors of multi-task learning, for related and unrelated tasks, and future work will have to address the two problems together.

MT-LRP is implemented as two neural networks, coupled by a *gating* mechanism (Sigaud et al., 2015; Droniou et al., 2015) as illustrated in Figure 2. The first network, $\chi$, detects which task is being executed and selects the corresponding state representation. The second network, $\varphi$, learns task-specific state representations. The networks are trained simultaneously using the *robotic priors* learning objective (Jonschkowski & Brock, 2015), exploiting physics-based prior knowledge about how states, actions, and rewards relate to each other. Both networks learn from raw sensor data, without supervision and solely based on the robot's experiences.

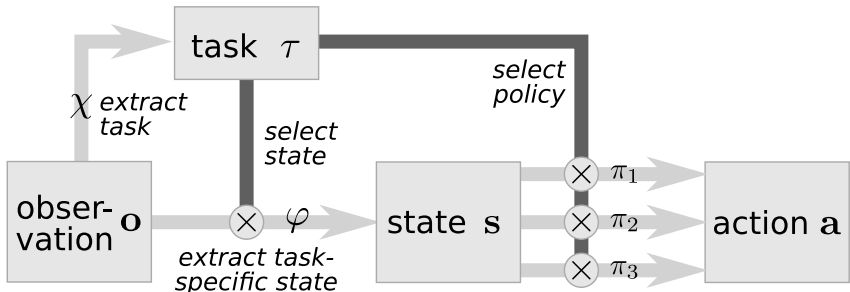

Figure 2: Overview of the gated network for state representation learning for multiple tasks.

In a simulated experimental scenario, we show that MT-LRP is able to learn multiple state representations and task detectors from raw observations and that these representations allow to learn better policies from fewer data when compared with other methods. Moreover, we analyze the contribution to this result of each the method's individual components.

## 2    RELATED WORK

MT-LRP combines three ideas into a novel approach for task discovery and state representation learning: 1) extracting state representations for each task with robotic priors (Jonschkowski & Brock, 2015); 2) discovering discrete tasks and corresponding actions/policies in a RL context (Stulp et al., 2014; Höfer & Brock, 2016); 3) using gated networks to implement a "mixture of experts" (Jacobs et al., 1991; Droniou et al., 2015).

**State Representation Learning**: Learning from raw observations is considered a holy grail in reinforcement learning (RL). Deep RL has had major success in this, using model-free (Mnih et al., 2015) but also by combining model-free and model-based RL (Levine et al., 2015). These approaches apply end-to-end learning to get from raw input to value functions and policies. A different approach is to explicitly learn state representations using unsupervised learning, e.g. using auto-encoders (Lange et al., 2012). Recently, Watter et al. (2015) extended this idea to learn state representations jointly with dynamic models and apply optimal control to compute a policy. We use learning with robotic priors (Jonschkowski & Brock, 2015), a state representation learning method

that exploits information about temporal structure, actions, and rewards. We go beyond previous work by not only learning single state representations, but learning multiple state representations given raw data from multiple tasks.

**Options and Parameterized Skills**: A common approach to factorizing a RL problem into subtasks are macro-actions, often called *options* (Sutton et al., 1999; Hengst, 2002). The main difference with our approach is that options are used to hierarchically decompose one high-level task into subtasks (and learn sub-policies for these subtasks), whereas we learn task-specific state representations for different high-level tasks. However, options bear resemblance on a technical level, since they are often implemented by a high-level "selection" policy that parametrizes low-level policies (Daniel et al., 2012; Kupcsik et al., 2013; Stulp et al., 2014). Continuous versions of options, referred to as parametrized skills, have been proposed, too (Da Silva et al., 2012; Deisenroth et al., 2014; Doshi-Velez & Konidaris, 2016). However, in all the work above, the state representation is given. To the best of our knowledge, state representation learning has not yet been considered in the context of RL with options or parameterized skills.

**Gated Networks for Mixtures of Experts and Submanifold Learning**: Gated networks are networks that contain gating connections, in which the outputs of at least two neurons are multiplied (Sigaud et al., 2015). This allows a *gating neuron g* to prohibit (or limit) the flow of information from one neuron *x* to another neuron *y*, similar to how transistors function. An early example of gated networks is the *mixture of experts* approach (Jacobs et al., 1991; Jacobs & Jordan, 1993; Haruno et al., 2001), where separate networks in a modular neural network specialize in predicting subsets of training examples from a database. Our contribution is to extend mixtures of experts by state representation learning (e.g. from raw images) and to the more difficult RL (rather than supervised learning) context. Our gated network architecture is similar to the one proposed by Droniou et al. (2015). Their network simultaneously learns discrete classes jointly with continuous class variations (called *submanifolds*) in an unsupervised way, e.g., discrete digit classes and shape variations within each class. We use a similar architecture, but in a different way: rather than learning discrete classes, we learn discrete tasks; class-specific submanifolds correspond to task-specific state representations; and finally, we consider a RL rather than an unsupervised learning context.

As mentioned in the introduction, our work is orthogonal to multi-task learning (Caruana, 1997) which has been extensively studied in recent reinforcement learning literature, too (Parisotto et al., 2016). Our approach can be trivially combined with multi-task learning by by prepending the gate and state extraction modules with a subnetwork that shares knowledge across tasks. Another interesting multi-task approach is policy distillation (Rusu et al., 2016). This method combines different policies for multiple tasks into a single network, which enables to share information between tasks and to learn a compact network that can even outperform the individual policies.

## 3 BACKGROUND: STATE REPRESENTATION LEARNING FOR REINFORCEMENT LEARNING

We formulate MT-LRP in a reinforcement learning (RL) setting using a Markov decision process (MDP) $(S, A, T, R, \gamma)$: Based on the current state $\mathbf{s} \in S$, the agent chooses and executes an action $a \in A$, obtains a new state $\mathbf{s}' \in S$ (according to the transition function $T$) and collects a reward $r \in R$. The agent's goal is to learn a policy $\pi : S \to A$ that maximizes the expected return $\mathrm{E}(\sum_{t=0}^{\infty} \gamma^t r_t)$, with $r_t$ being the reward collected at time $t$ and $0 < \gamma \le 1$ the discount factor. We consider an episodic setting with episodes of finite length, a continuous state space $S$ and a discrete action space $A$.

In this work, we assume that the agent *cannot* directly observe the state $\mathbf{s}$ but only has access to observations $\mathbf{o} \in O$, which are usually high-dimensional and contain task-irrelevant distractors. This requires us to extract the state from the observations by learning an observation-state-mapping $\varphi : O \to S$, and use the resulting state representation $S$ to solve the RL problem (assuming that a Markov state can be extracted from a single observation). To learn the state representation, we apply *learning with robotic priors* (Jonschkowski & Brock (2015), from now on referred to as *LRP*). This method learns $\varphi$ from a set of temporally ordered experiences $D = \{(\mathbf{o}_t, a_t, r_t)\}_{t=1}^d$ by optimizing the following loss:

$$\mathcal{L}_{\mathrm{RP}}(D, \varphi) = \omega_t \mathcal{L}_{\mathrm{temp.}}(D, \varphi) + \omega_p \mathcal{L}_{\mathrm{prop.}}(D, \varphi) + \omega_c \mathcal{L}_{\mathrm{caus.}}(D, \varphi) + \omega_r \mathcal{L}_{\mathrm{rep.}}(D, \varphi). \tag{1}$$

This loss consists of four terms, each expressing a different prior about suitable state representations for robot RL. We optimize it using gradient descent, assuming $\varphi$ to be differentiable. We now explain the four robotic prior loss terms in Eq. (1).

*Temporal Coherence* enforces states to change gradually over time (Wiskott & Sejnowski, 2002):

$$\mathcal{L}_{\text{temp.}}(D, \varphi) = \mathbf{E}\left[\|\Delta \mathbf{s}_t\|^2\right],$$

where $\Delta \mathbf{s}_t = \mathbf{s}_{t+1} - \mathbf{s}_t$ denotes the state change. (To increase readability we replace $\varphi(\mathbf{o})$ by $\mathbf{s}$.) *Proportionality* expresses the prior that the same action should change the state by the same magnitude, irrespective of time and the location in the state space:

$$\mathcal{L}_{\text{prop.}}(D, \varphi) = \mathbf{E}\left[(\|\Delta \mathbf{s}_{t_2}\| - \|\Delta \mathbf{s}_{t_1}\|)^2 \mid a_{t_1} = a_{t_2}\right].$$

*Causality* enforces two states $\mathbf{s}_{t_1}, \mathbf{s}_{t_2}$ to be dissimilar if executing the same action in $\mathbf{s}_{t_1}$ generates a different reward than in $\mathbf{s}_{t_2}$.

$$\mathcal{L}_{\text{caus.}}(D, \varphi) = \mathbf{E}\left[e^{-\|\mathbf{s}_{t_2} - \mathbf{s}_{t_1}\|^2} \mid a_{t_1} = a_{t_2}, r_{t_1+1} \neq r_{t_2+1}\right].$$

*Repeatability* requires actions to have repeatable effects by enforcing that the same action produces a similar state change in similar states:

$$\mathcal{L}_{\text{rep.}}(D, \hat{\varphi}) = \mathbf{E}\left[e^{-\|\mathbf{s}_{t_2} - \mathbf{s}_{t_1}\|^2} \|\Delta \mathbf{s}_{t_2} - \Delta \mathbf{s}_{t_1}\|^2 \mid a_{t_1} = a_{t_2}\right].$$

Additionally, the method enforces *simplicity* by requiring $\mathbf{s}$ to be low-dimensional.

Note that learning with robotic priors only makes use of the actions $a$, rewards $r$, and temporal information $t$ during optimization, but not at test time for computing $\varphi(\mathbf{o}) = \mathbf{s}$. Using $a$, $r$ and $t$ in this way is an instance of the learning with side information paradigm (Jonschkowski et al., 2015).

# 4 MULTI-TASK STATE REPRESENTATIONS: MT-LRP

Now consider a scenario in which an agent is learning multiple distinct tasks. For each task $\tau \in \{1, \ldots, T\}$, the agent now requires a task-specific policy $\pi_\tau : S_\tau \to A$. We approach the problem by learning a task-specific *state representation* $\varphi_\tau : O \to S_\tau$ for each policy, and a *task detector* $\chi$ which determines the task, given the current observation. We will consider a probabilistic task-detector $\chi : O \to [0,1]^T$ that assigns a probability to each task being active.

In order to solve the full multi-task RL problem, we must learn $\chi$, $\{\varphi_\tau\}_{\tau \in \{1,\ldots,T\}}$ and $\{\pi_\tau\}_{\tau \in \{1,\ldots,T\}}$. We propose to address this problem by MT-LRP, a method that jointly learns $\chi$ and $\{\varphi_\tau\}_{\tau \in \{1,\ldots,T\}}$ from raw observations, actions, and rewards. MT-LRP then uses the state representations $\{\varphi_\tau\}$ to learn task-specific policies $\{\pi_\tau\}_{\tau \in \{1,\ldots,T\}}$ (using standard RL methods), and switches between them using the task detector $\chi$. To solve the joint learning problem, MT-LRP generalizes LRP (Jonschkowski & Brock, 2015) in the following regards: (i) we replace the linear observation-state-mapping from the original method with a *gated neural network*, where the gates act as task detectors that switch between different task-specific observation-state-mappings; (ii) we extend the list of robotic priors by the prior of *task coherence*, which allows us to train multiple task-specific state representations without any specification (or labels) of tasks and states.

## 4.1 GATED NEURAL NETWORK ARCHITECTURE

We use a gated neural network architecture as shown schematically in Fig. 2. The key idea is that both the task detector $\chi$ as well as the state representation $\varphi$ are computed from raw inputs. However, the output of the task detector *gates* the output of the state representation. Effectively, this means the output of $\chi(\mathbf{o})$ decides which task-specific state representation $\varphi_\tau$ is passed further to the policy, which is also gated by the output of $\chi(\mathbf{o})$.

Formally, $\chi(\mathbf{o}) = \sigma(\chi_{\text{pre}}(\mathbf{o}))$ is composed of a function $\chi_{\text{pre}}$ with $T$-dimensional output and a *softmax* $\sigma(\mathbf{z}) = \frac{e^{z_j}}{\sum_k e^{z_k}}$. The softmax ensures that $\chi$ computes a proper probability distribution over tasks. The probabilities are then used to gate $\varphi$. To do this, we decompose $\varphi$ into a *pre-gating function*

$\varphi_{\text{pre}}$ that extracts features shared across all tasks (i.e. "multi-task" in the sense of Caruana (1997), unless set to the identity), and a $T \times M \times N$ *gating tensor* **G** that encodes the $T$ (linear) observation-state mappings ($M = \dim(\mathbf{s})$ and $N$ is the output dimension of $\varphi_{\text{pre}}$). The value of the state's $i$-th dimension $\mathbf{s}_i$ computes as the expectation of the dot product of gating tensor and $\varphi_{\text{pre}}(\mathbf{o})$ over the task probabilities $\chi(\mathbf{o})$:

$$s_i = \varphi_i(\mathbf{o}) = \sum_{k=1}^{T} \chi_k(\mathbf{o}) \langle \mathbf{G}_{k,i,:}, \varphi_{\text{pre}}(\mathbf{o}) \rangle. \tag{2}$$

## 4.2 LEARNING OBJECTIVE

To train the network, we extend the robotic prior loss $\mathcal{L}_{\text{RP}}$ (Eq. 1), by a *task-coherence prior* $\mathcal{L}_\tau$:

$$\mathcal{L} = \mathcal{L}_{\text{RP}}(D, \varphi) + \omega_\tau \mathcal{L}_\tau(D, \chi), \tag{3}$$

where $\omega_\tau$ is a scalar weight balancing the influence of the additional loss term. Task coherence is the assumption that a task only changes between training episodes, not within the same episode. It does not presuppose any knowledge about the number of tasks or the task presented in an episode, but it exploits the fact that task switching weakly correlates with training episodes. Moreover, this assumption only needs to hold during training: since $\chi$ operates directly on the observation $\mathbf{o}$, it can in principle switch the task at every point in time during execution. Task-coherence applies directly to the output of the task detector, $\chi(\mathbf{o})$, and consists of two terms:

$$\mathcal{L}_\tau^{\text{con+sep}} = \mathcal{L}_\tau^{\text{con}} + \mathcal{L}_\tau^{\text{sep}}. \tag{4}$$

The first term enforces *task consistency* during an episode:

$$\mathcal{L}_\tau^{\text{con}} = \mathbf{E}\Big[ H(\chi(\mathbf{o}_{t_1}), \chi(\mathbf{o}_{t_2})) \;\Big|\; \text{episode}_{t_1} = \text{episode}_{t_2} \Big], \tag{5}$$

where $H$ denotes the cross-entropy $H(p,q) = -\sum_x p(x) \log q(x)$. It can be viewed as a measure of dissimilarity between probability distributions $p$ and $q$. We use it to penalize $\chi$ if it assigns different task distributions to inputs $\mathbf{o}_{t_1}$, $\mathbf{o}_{t_2}$ that belong to the same episode. Note that task-consistency can be viewed as a temporal coherence prior on the task level (Wiskott & Sejnowski, 2002).

The second term expresses *task separation* and encourages $\chi$ to assign tasks to different episodes:

$$\mathcal{L}_\tau^{\text{sep}} = \mathbf{E}\Big[ e^{-H(\chi(\mathbf{o}_{t_1}), \chi(\mathbf{o}_{t_2}))} \;\Big|\; \text{episode}_{t_1} \neq \text{episode}_{t_2} \Big]. \tag{6}$$

This loss is complementary to task consistency, as it penalizes $\chi$ if it assigns similar task distributions to $\mathbf{o}_{t_1}$, $\mathbf{o}_{t_2}$ from different episodes. Note that $\mathcal{L}_\tau^{\text{sep}}$ will in general *not* become zero. The reason is that the number of episodes usually exceeds the number of tasks, and therefore two observations from different episodes sometimes do belong to the same task. We will evaluate the contribution of each of the two terms to learning success in Section 5.2.

## 5 EXPERIMENTS

We evaluate MT-LRP in two scenarios. In the *multi-task slot-car racing* scenario (inspired by Lange et al. (2012)), we apply MT-LRP to a linearly solvable problem, allowing us to easily inspect what and how MT-LRP learns. In slot-car racing, the agent controls one of multiple cars (Figure 1), with the goal of traversing the circuit as fast as possible without leaving the track due to speeding in curves. However, the agent does not know a priori which car it controls, and only receives the raw visual signal as input. Additionally, uncontrolled cars driving at random velocity, act as visual distractors. We turn this scenario into a multi-task problem in which the agent must learn to control *each* car, where controlling the different cars corresponds to separate tasks. We will now provide the technical details of our experimental set-up.

### 5.1 EXPERIMENTAL SET-UP: SLOT-CAR RACING

The agent controls the velocity of one car (see Fig. 1), receives a reward proportional to the car's velocity, chosen from $[0.01, 0.02, \ldots, 0.1]$, and a negative reward of $-10$ if the car goes too fast

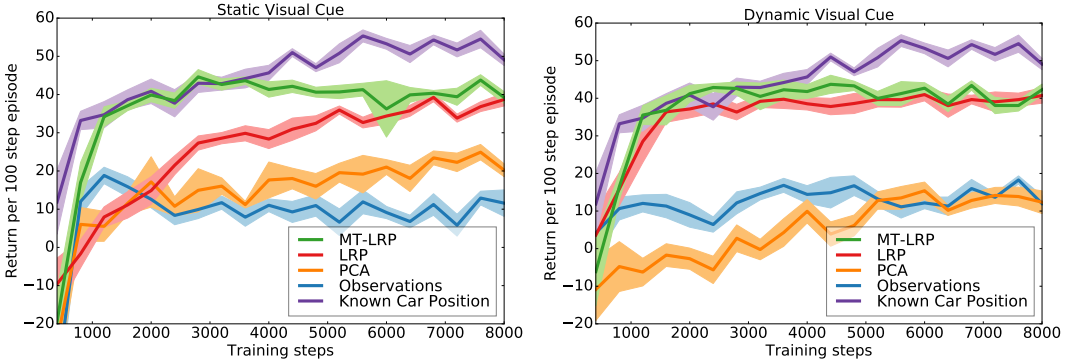

Figure 3: Reinforcement learning curves (mean and standard error) for different state representations for the *two*-slot car scenarios. Left: static visual cue. Right: dynamic visual cue.

in curves. The velocity is subject to Gaussian noise (zero mean, standard deviation 10%) of the commanded velocity. All cars move on independent lanes and do not influence each other. The agent observes the scenario by getting a downscaled 16x16 RGB top-down view (dimension $N = 16 \times 16 \times 3 = 768$) of the car circuit (Fig. 1(b)).

In our experiments, there are two or three cars on the track, and the agent controls a different one in every episode. To recognize the task, the agent must be able to extract a visual cue from the observation which correlates with the task. We study two types of visual cues:
*Static Visual Cue:* The arrangement of cars stays the same in all episodes and a static visual cue (a picture of the controlled car) in the top-left image corner indicates which car is currently controlled.
*Dynamic Visual Cue:* The agent always controls the same car (with a certain color), but in each task the car is located on a different lane (as in Fig. 1(b)).

**Data Collection and Learning Procedure:** The agent collects 40 episodes per task, each episode consisting of 100 steps. To select an action in each step, the agent performs $\varepsilon$-greedy exploration by picking a random action with probability $\varepsilon = 0.3$ and the best action according to its current policy otherwise. The agent computes a policy after every $\tau$ episodes, by first learning the observation-state mapping $\varphi$ (state representation) and then computing policies $\pi_1, \dots, \pi_\tau$ (based on the outcomes of the learned $\chi$ and $\varphi$). To monitor the agent's learning progress, we measure the average reward the agent attains on $T$ test episodes, i.e. one test episode of length 100 per task (using the greedy policy), amounting to 8000 experiences in total. To collect sufficient statistics, the whole experiment is repeated 10 times.
**Policy Learning:** We consider the model-free setting with continuous states $S$, discrete actions $A$ and solve it using nearest-neighbor Q-learning *kNN-TD-RL* (Martín H et al., 2009) with $k = 10$. More recent approaches to model-free RL would be equally applicable (Mnih et al., 2015).
**Learning Strategies and Baselines:** We compare five strategies. We run a) MT-LRP with 5 gate units (two/three more than necessary), state dimensionality $M = 2$ and using $\mathcal{L}_\tau^{\text{con+sep}}$ as task-coherence prior. We compare MT-LRP to several state representation methods; for each method we evaluate different $M$ and report only the best performing $M$: a) robotic priors without gated network, LRP ($M = 4$), b) principal components analysis (*PCA*) on the observations ($M = 20$) and c) raw observations ($M = 768$). Additionally, we evaluate d) a lower baseline in the form of a randomly moving agent and e) an upper baseline by applying RL on the known 2D-position of the slot car under control ($M = 2$). We use the same RL algorithm for all methods. To learn the state representations with robotic priors, we base our implementation on Theano and lasagne, using the Adam optimizer with learning rate 0.005, batch size 100, Glorot's weight initialization and $\omega_t = 1, \omega_p = 5, \omega_c = 1, \omega_r = 5, \omega_\tau = 10$. Moreover, we apply an L1 regularization of 0.001 on $\varphi$.
Additionally, we analyze the contribution of task coherence priors by applying MT-LRP to the full set of 8000 experiences a) without task-coherence, b) with task consistency $\mathcal{L}_\tau^{\text{con}}$ only c) with task separation $\mathcal{L}_\tau^{\text{con}}$ only) and d) without task consistency and separation $\mathcal{L}_\tau^{\text{con+sep}}$.

## 5.2 RESULTS

We will now present the three main results of our experiments: (i) we show that MT-LRP enables the agent to extract better representations for RL; (ii) we provide insight in how the learner detects the task and encodes the state representations; and finally, (iii) we show the contribution of each of the task-coherence loss terms.

**MT-LRP Extracts Better State Representations for RL**  Figure 3 shows the learning curves for RL based on state representations learned by the different methods in the two-slot-car scenario (static visual cue on the left, dynamic on the right). No method reaches the performance of the upper baseline, mainly due to aliasing errors resulting from the low image resolution.

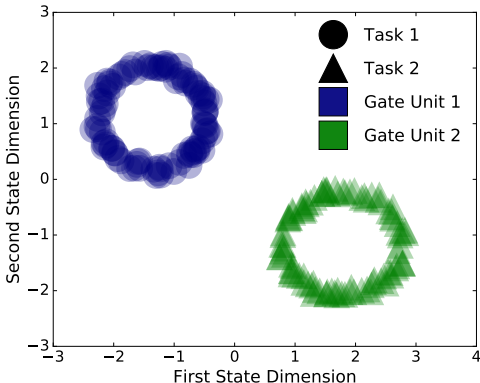

The random baseline ranges around an average reward of $-84.9$ with standard error 0.72 and was omitted from the Figure. The state representation learning baselines without robotic priors perform poorly because they are unable to identify the task-irrelevant distractions. MT-LRP gets very close to the performance of the upper baseline, especially for very low amounts of training data ($d < 2500$), whereas LRP does not even attain this level of performance for the full training set $d = 8000$ in the static task. The gap between MT-LRP and LRP increases even more if we add another car (Figure 5) because LRP can only learn one state representation for all three tasks. Including the three slot cars in this representation results in distractions for the RL method. However, in the dynamic-visual-cue scenario LRP-4 performs on par with MT-LRP. Surprisingly, running LRP with only *two* dimensions suffices to achieve the performance of MT-LRP. We

Figure 4: State representation learned per task (different markers) and per gate unit (different colors)

will explain this phenomenon below. To conclude, MT-LRP allows to learn as good or better policies than the baselines in all slot-car scenarios.

**MT-LRP Detects All Tasks and Learns Good State Representations**  To gain more insight into what is learned, we analyze the state representations extracted by MT-LRP and LRP. Figure 4 shows the state representation learned by MT-LRP for the static-visual-cue scenario. Each point in the figure corresponds to one observation, markers indicate the task and colors the most active gate unit. We see that the first gate unit (blue) is always active for task 1 (circle), and the second gate unit for task 2. This shows that the task is detected with high accuracy. The task detector $\chi$ is also highly cer-

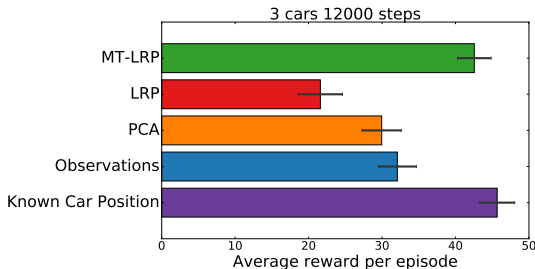

Figure 5: Reinforcement learning performance in the *three*-slot car scenario with static visual cue.

tain which is reflected in the fact that its entropy evaluated on the data is close to zero. Moreover, the states reflect the circular structure of the slot car racing track. We thus conclude that MT-LRP has learned to identify the tasks and to represent the position of each car on the track.

The RL experiments raised the question why LRP manages to solve the dynamic, but not the static-visual-cue scenario as well as MT-LRP. We hypothesize that, for the dynamic cue, LRP is able to extract the position of the car on regardless of which lane it is in using a single linear mapping. Figure 6 confirms this hypothesis: LRP filters for the car's color (blue) along the track and assigns increasing weights to these pixels which results in the extraction of its position. It also assigns constant weights along the track in the red channel using the lane change of the two cars as an offset. This results in a mapping to two circles similar to Fig. 4, where the state encodes both the position and the task. Such a mapping can be expressed by a linear

function precisely because the features that are relevant for one task do not reappear in another task (e.g. a blue slot car in track 1 does not appear in the task where the blue car is in track 2).

However, there exists no equivalent linear mapping for the static-visual-cue variant of the slot-car problem, because cars that are relevant for one task are also present in every other task.

We can generalize from this insight as follows. A single linear observation-state-mapping is sufficient for multiple tasks if the state representation for every task can be extracted by a linear function using only features that stay constant for all other tasks. If this is the case, than there is no need for decoupling the extraction of task and state.

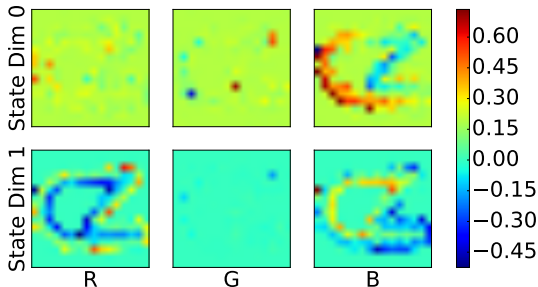

Figure 6: $\varphi$ learned by LRP ($M = 2$) for the two-car dynamic visual cue tasks. Row corresponds to state dimension, column to RGB color channel.

**Task-Consistency is Critical for Learning Performance**  To understand the influence of the different task-coherence prior variants, we compared their performance in Figure 7. We see that relying solely on the robotic priors gives poor results, mainly because the gate units are not used properly: more than one gate unit is activated per task ($\chi$ has high entropy). Adding the task-separation prior forces the network to use as many gates as possible (5 in our case), leading to bad state representations. Interestingly, using task consistency only gives roughly the same result as using task consistency and task separation.

**Discussion**  The experiments showed that MT-LRP is able to solve the representation and reinforcement learning tasks better than the baselines. Important questions for future work concern: the necessity and influence of the task-separation loss, in particular for short episode lengths and if the number of expected tasks exceeds the number of actual tasks; and transferring knowledge by adding a shared neural network layers before gating.

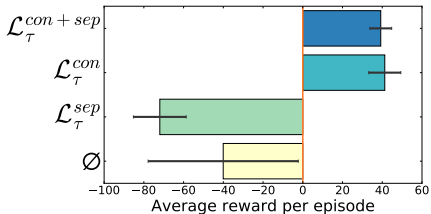

Figure 7: Task coherence: Average reward per episode (8000 samples).

## 6  CONCLUSION

We have presented MT-LRP, a method for multi-task state representation learning with robotic priors. The method learns in an unsupervised fashion, solely based on the robots own observations, actions, and rewards. Our experiments confirmed that MT-LRP is effective in simultaneously identifying tasks and learning task-specific state representations. This capability is beneficial for scaling reinforcement learning to realistic scenarios that require dedicated skills for different tasks.

ACKNOWLEDGMENTS

We gratefully acknowledge the funding provided by the German Research Foundation (DFG, Exploration Challenge, BR 2248/3-1), the Alexander von Humboldt foundation through an Alexander von Humboldt professorship (funded by the German Federal Ministry of Education and Research). Additionally, Antonin Raffin was supported by an Erasmus+ grant.

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
