# Peer review of "Unsupervised Learning of State Representations for Multiple Tasks"

_ICLR 2017 — rejected_

[Official Review · AnonReviewer3 · rating 6 · confidence 4 · 16 Dec 2016]
**Unsupervised Learning of State Representations for Multiple Tasks**

This paper is about learning unsupervised state representations using multi-task reinforcement learning.  The authors propose a novel approach combining gated neural networks with multitask learning with robotics priors. They evaluated their approach on two simulated datasets and showed promising results. The paper is clearly written and is theoretically sound.

Positives:
+ Gating to enable learning a joint representation
+ Multi-task learning extended from a single task in prior work
+ Combining multiple types of losses to learn a strong representation (Coherence, Proportionality, Causality, Repeatability, Consistency and Separation)

Negatives:
- Parameters choice is arbitrary (w parameters)
- Limiting the multi-task learning to be different to individual tasks rather than sharing and transferring knowledge between tasks
- The experiments could have been conducted using a standardized simulation tool such as OpenAI Gym to make it easy to compare.

I would recommend that the authors consider a more standardized way of picking the model parameters and evaluate on a more standard and high-dimensional datasets.

[Official Review · AnonReviewer1 · rating 5 · confidence 4 · 16 Dec 2016]

This paper builds upon the method of Jonschkowski & Brock to learn state representations for multiple tasks, rather than a single task. The research direction of learning representations for multiple tasks is an interesting one, and largely unexplored. The approach in the paper is to learn a different representation for each task, and a different policy for each task, where the task is detected automatically and built into the neural network.

The authors state that the proposed method is orthogonal to multi-task learning, though the end goal of learning to solve multiple tasks is the same. It would be interesting and helpful to see more discussion on this point in the paper, as discussed in the pre-review question phase. References to other multi-task learning works, e.g. policy distillation and actor-mimic (both ICLR ’16), may be appropriate as well.

The method proposes to jointly learn a task classifier with a state representation learner, by using a differentiable gating mechanism to control the flow of information. The paper proposes a task coherence prior for this gating mechanism to ensure that the learned task classifier is temporally coherent. Introducing this structure is what enables the method to improve performance over the standard, non-multitask approach.

The evaluation involves two toy experimental scenarios. The first involves controlling one of two cars to drive around a track. In this task, detecting the “task” is very easy, and the learned state representation is linear in the observation. The paper evaluates the performance of the policies learned with the proposed approach, and shows sufficient comparisons to demonstrate the usefulness of the approach over a standard non-multitask set-up.

In the second navigation scenario, only the state representation is qualitatively shown, not the resulting control policy nor any other learned state representations for comparison. Since the multi-task state representation learning approach is only useful if you can also learn control better, the paper should also evaluate on control, with the same comparisons as in the first experiment. Without this evaluation, the experiment is incomplete.

Lastly, to be on par with publications at a venue like ICLR, the method should be evaluated more thoroughly, on a wider range of set-ups, to demonstrate the generality of the approach and show that the method applies to more complex tasks. While in theory, the method should scale, the experiments do not demonstrate that it can handle more realistic scenarios, such as scaling beyond MNIST-level images, to 3D or real images, or higher-dimensional control tasks. Evaluating the method in this more complex scenario is important, because unexpected issues can come up when trying to scale. If scaling-up is straight-forward, then running this experiment (and including it in the paper) should be straight-forward.

In summary, here are the pros and cons of this paper:
Cons
- The approach does not necessarily share information across tasks for better learning, and requires learning a different policy for each task
- Only one experimental set-up that evaluates learned policy with multi-task state representation
- No experiments on more realistic scenarios, such 3D environments or high-dimensional control problems
Pros: 
- This approach enables using the same network for multiple tasks, which is often not true for transfer and multi-task learning approaches
- Novel way to learn a single policy for multiple tasks, including a task coherence prior which ensures that the task classification is meaningful
- Experimentally validated on two toy tasks. One task shows improvement over baseline approaches

Thus, my rating would be higher if the paper included an evaluation of the control policy for navigation and included another more challenging and compelling scenario.


Lastly, here are some minor comments/questions on how I think the paper could be improved, but are not as important as the above:

Approach:
Could this approach be combined with other state representation learning approaches? e.g. approaches that use an autoencoder.

Experiments:
One additional useful comparison would be to evaluate performance in the single-task setting (e.g. only controlling the red car), as an upper bound on how well the policy should be able to perform. Does the learned multi-task policy reach the same level of performance? This upper bound will be tighter than the “known car position” baseline (which is also useful in its own right).

Does the “observations” baseline eventually reach the performance of the LRP approach? It would be useful to know if this approach simply speeds up learning (significantly) or if it enables better performance.

If there are aliasing issues with the images, why not just use higher resolution images?

[Official Review · AnonReviewer2 · rating 6 · confidence 3 · 16 Dec 2016]
**Unsupervised Learning of State Representations for Multiple Tasks**

The paper presents a method to learn a low-dimensional state representations from raw obervation for multi-task setting. In contrast to classic multi-task learning setting where a joint representation is usually learned by exploring the transferable information among different tasks, the method aims to identify individual task and solve them separately. To this end, the authors extend the learning with robotic priors approach by extending the loss function with additional term for task coherence, i.e., a task only changes representation between training episodes. The method has been evaluated on two tasks, multi-task slot-car racing and mobile navigation to prove its efficacy.

there were several unclear issues:

1. The first question is that if the method is only appealing on the scenario like the slot-car racing, otherwise it should be benchmarked with mutli-task learning. While the author made the argument in the related work, the proposed method is orthogonal to multi-task learning they did admit both explore shared knowledge between tasks. What's the advantage and disadvantage for the proposed method for general mutiple task setting, in particular over the multi-task learning?
The reply of the authors was not fully satisfactory. The argument did not support the lack of comparison to multi-task joint-learning. It seems they don't plan to include any comparison neither. I think it's important for the fundamental motivation for the work, without such comparison, the method seems to be purely an alternative to multi-task joint-learning without any(or much) practical advantage.

2.Following up to the previous question, please clarify the results on the mobile navigation scenario. It's not clear how the plot on the right indicates MT-LRP identifies all tasks as the author claimed and and seems very weak to support the method, in particular compared to the multi-slot car-racing driving experiment, there is too little results to make sound argument (almost no comparison to alternative methods, i.e. no baseline method, is that true for the problem).
The explanation of the authors did provide more details and more explicit information. 

3. The proposed gated neural network architecture seems to be a soft gated structure(correct me if I am wrong), a possible baseline would be a hard gated unit, how would this affect the conclusion. This is particularly interesting as the authors reflect on the constraint that the representation should stay consistent during the training.
The author simply stated again what they did for the modeling without counter the comparison to hard-gating, but it's probably less an issue compared to Question 1.

In summary, while there are remaining concerns about lacking comparisons, the is a weak tendency towards accepting the submission.

[Author Response · Sebastian Höfer · 13 Jan 2017]
**Reply to Reviewer Comments**

We would like to thank all reviewers for their thorough and helpful comments!

1) Before we turn to the individual questions raised the reviewers, we would like to address the main issue that all reviewers raised, namely the relationship of our method to multi-task learning:

“The authors state that the proposed method is orthogonal to multi-task learning though the end goal of learning to solve multiple tasks is the same.” (AnonReviewer1)
“The argument did not support the lack of comparison to multi-task joint-learning.” (AnonReviewer2)
“Limiting the multi-task learning to be different to individual tasks rather than sharing and transferring knowledge between tasks” (AnonReviewer 3)

The intro has been rewritten to clarify our motivation and how our work compares to multi-task learning. We completely agree that successful RL will require multi-task learning to share knowledge that generalizes over multiple tasks. 

But there are sets of tasks that require multiple dedicated skills without sharing knowledge. For instance, in a video game, an agent have to achieve several subgoals (fight an enemy, avoid obstacles,...), each of these can be seen as individual task. Learning multiple, (sub-)policies dedicated to *different* tasks is a problem of its own right, as it faces significant theoretical issues, such as “catastrophic forgetting”. We have elaborated on this argument in the introduction.

Since there is few work approaching this problem in RL, our paper studies the question of how to learn fully independent policies for different tasks. We fully agree that future work will need to combine learning shared and separate representations but we regard our work on the  independent-policy multi-task RL problem as a contribution in itself. 

We now reply to the individual comments raised by the reviewers.

----

AnonReviewer1

1) “References to other multi-task learning works, e.g. policy distillation and actor-mimic (both ICLR ’16), may be appropriate as well.”

Thank you for the pointers, we have integrated the two suggested papers in the related work of the paper.

2) “The approach does not necessarily share information across tasks for better learning, and requires learning a different policy for each task”

This is the very idea of the method proposed, we updated the introduction to clarify the reasons we focused on this approach.

3) “In the second navigation scenario, only the state representation is qualitatively shown, not the resulting control policy nor any other learned state representations for comparison. Since the multi-task state representation learning approach is only useful if you can also learn control better, the paper should also evaluate on control, with the same comparisons as in the first experiment. Without this evaluation, the experiment is incomplete.”

We agree and as mentioned before this was a preliminary and incomplete experiment, and we decided to remove it from the paper. 

4) “Lastly, to be on par with publications at a venue like ICLR, the method should be evaluated more thoroughly, on a wider range of set-ups [...]”

We agree that it is beneficial to apply a method to a wider range of tasks. Yet, we chose to invest into rigorously evaluating the performance of the method on the chosen task, and provide a thorough argument why and how the method works. We believe that it will scale to a wider range of tasks, but we will have to address this in future work.

5) “Could this approach be combined with other state representation learning approaches? e.g. approaches that use an autoencoder.”

Yes, in principle it would be possible to use other state representation learning objectives.  Note, however, that in the slot car racing scenario a PCA/auto-encoder loss will not perform as well as LRP, as it has will try to explain all variations in the observation, in particular the second slot car. This has been shown in our previous work (Jonschkowski & Brock 2015) and is also reflected in the performance of PCA in the slot-car experiment.

6) ”One additional useful comparison would be to evaluate performance in the single-task setting (e.g. only controlling the red car), as an upper bound on how well the policy should be able to perform. Does the learned multi-task policy reach the same level of performance? This upper bound will be tighter than the “known car position” baseline (which is also useful in its own right).”

Thank you for this suggestion; in our experiment, however, the performance of the car in the single-task setting is identical to the performance we see in the multi-task setting. The reason is that the task detector module has a very high accuracy (greater than 99%) for the slot car tasks, and in consequence, a separate policy for each slot car is learned.

7) “Does the “observations” baseline eventually reach the performance of the LRP approach? It would be useful to know if this approach simply speeds up learning (significantly) or if it enables better performance.”

Our experiments and previous work (Jonschkowski & Brock 2015) suggest that it will eventually reach the same performance with enough data, but for now, even in our largest experiments, we did not see it happening.

8) “If there are aliasing issues with the images, why not just use higher resolution images?”

Mainly computational reasons: we wanted to evaluate a wide variety of parameter settings and study their influence on our algorithm, yet we did not have the computational power to do this exhaustively on higher resolutions.

---

AnonReviewer2

1) “The author simply stated again what they did for the modeling without counter the comparison to hard-gating, but it's probably less an issue compared to Question 1.”

We are sorry that our answer in the pre-review phase did not address your question. We were trying to explain that our method is technically a soft gating, but effectively learns to perform hard gating. We are not sure how whether and how using a hard would influence the conclusion of the paper, and we are not aware of a way to implement a differentiable hard gate (if it is not differentiable, we cannot train it using backpropagation).

---

AnonReviewer3

2) “Parameters choice is arbitrary (w parameters)”
The weights w for the different methods are chosen as described in Jonschkowski & Brock 2015, by monitoring the gradient on a small part of the training set. The goal is to have gradients of the same magnitude for the different terms in the loss, so only relative weighting matters. Small changes to the parameters do not affect the method and there is no need for careful tuning.

3) “The experiments could have been conducted using a standardized simulation tool such as OpenAI Gym to make it easy to compare.”
We agree that evaluating experiments on a standardized tool such as OpenAI gym is a great idea. We want to point out, though, that the slot car racing task considered in the paper is a well-known task that has been evaluated in previous work, too, e.g. (Lange et al., 2012). Moreover, it is the simplest task that has the properties we are interested in this paper (non-overlapping tasks).
But we agree that that open simulation tools such as OpenAI gym are great and we will apply our method to these tasks in future work.

[Final Decision · Program Chairs · 06 Feb 2017]
**ICLR committee final decision**

The authors propose to explore an important problem -- learning state representations for reinforcement learning. However, the experimental evaluation raised a number of concerns among the reviewers. The method is tested on only a single relatively easy domain, with some arbitrarily choices justified in unconvincing ways (e.g. computational limitations as a reason to not fix aliasing). The evaluation also compares to extremely weak baselines. In the end, there is insufficient evidence to convincingly show that the method actually works, and since the contribution is entirely empirical (as noted by the reviewers in regard to some arbitrary parameter choices), the unconvincing evaluation makes this method unsuitable for publication at this time. The authors would be encouraged to evaluate the method rigorously on a wide range of realistic tasks.